# Bridging the Action Gap by Democratizing Climate Change Education—The Case of *k.i.d.Z.21* in the Context of Fridays for Future

**Veronika Deisenrieder \***, **Susanne Kubisch, Lars Keller** and **Johann Stötter**

Institute of Geography, University of Innsbruck, 6020 Innsbruck, Austria; susanne.kubisch@uibk.ac.at (S.K.); lars.keller@uibk.ac.at (L.K.); johann.stoetter@uibk.ac.at (J.S.)

\* Correspondence: veronika.deisenrieder@uibk.ac.at; Tel.: +43-512-507-54040

**Abstract:** Formal schooling frequently lacks both democratic learning culture and effective climate change education (CCE). This study analyzes the effects of the participatory CCE initiative *k.i.d.Z.21* and the impacts of the current Fridays For Future (FFF) climate protests on teenagers' climate change awareness. The mixed-methods approach comprises online pre-and post-tests, and personal interviews with selected students. *k.i.d.Z.21* follows moderate constructivist, inquiry-based learning approaches and addresses 14-year old students from secondary schools in Austria and southern Germany. Considering the effectiveness of the CCE intervention of school year 2018/2019 (N = 169), quantitative findings reveal an increased mean of major components of climate change awareness, including climate-friendly behavior. When separating participants and non-participants in Fridays For Future, personal concern and refusing meat have both increased significantly only among protest participants. A closer examination of this group identifies an enhanced feeling of self-efficacy that might be triggered by perceived collective efficacy. Besides, more climate-friendly consumption behavior, as well as enhanced multiplicative action, are detected. The interviewed students also clearly assigned increased action-related components of climate change awareness to the attendance of FFF. From the findings, we conclude that democratic learning in and out of school can enhance action-related components of climate change awareness, and a combination of both can have an even stronger effect.

**Keywords:** climate-friendly-action; self-efficacy; CCE; Fridays For Future; democratic

## 1. Introduction

By the end of 2019, the 25th conference of the parties has again failed to deliver clear regulations to fulfill the promises from the Paris Agreement of 2015. Safeguarding human existence within the boundaries of the earth system, however, demands uncompromised collective action plus a change of individual behavior. As today's young people have to deal with the consequences of climate change more than any generation before, they are expected to be most heavily concerned about a livable future, which is a vital precondition for taking action [1]. In strengthening other components of young people's climate change awareness (CCA), climate change education becomes a critical factor [2,3]. Thus, educators have to empower the decision-makers of tomorrow with effective educational settings.

However, focusing particularly on the knowledge-awareness-action gaps [4,5], designing effective CCE-learning settings is not about a sheer transfer of knowledge far off from students personal interests, and often coined by teacher-centered educational styles [6,7]. However, traditional learning approaches are still dominant, which is also true for CCE [8]. Yet the complex social and ecological dimensions deriving from changing climate conditions call for a vigorous and innovative educational approach [9,10].

For achieving desired climate change related action, educational approaches granting a high degree of autonomy in determining both content and methodology by active participation of learners become relevant [11–13]. Here, moderate constructivism is among the most commonly applied learning theories. Following these approaches, the Austrian climate change education initiative, *k.i.d.Z.21*, aims at creating effective learning settings in order to influence and change various components of young peoples' CCA, ultimately in order to trigger climate-friendly action [14].

As in early 2019, climate school strikes have also started in Austria [15]; the potential effect of FFF on the *k.i.d.Z.21* learning intervention objectives is of interest to this study. This becomes especially relevant as political engagement holds the potential for triggering self-efficacy, a CCA-component that enables individuals to dismantle barriers to action in private life [16]. Thus, based on a higher level of collective efficacy, a higher degree of environmentally-friendly behavior among attendants of FFF is also hypothesized [17,18]. Therefore, this study investigates if single components of CCA differentiate between *k.i.d.Z.21*-participants who have and those who have not been involved in FFF.

This article is divided into six main sections. First, a theoretical background of CCA in the context of CCE and the knowledge–awareness–action gap is presented. Then, the educational setting of the applied CCE *k.i.d.Z.21* is detailed in the context of democratizing learning settings. Next, a link from FFF to CCA is given. In the subsequent chapter, the effectiveness of the *k.i.d.Z.21* program regarding its aims of enhancing defined components of CCA is evaluated, including action-related components, such as self-efficacy and the ability to act environmentally-friendly. Thereby, FFF participants are treated as a separate group in order to identify differences in their respective CCA components. Subsequently, the findings are complemented by answers resulting from personal interviews with ten selected students who participated both in the *k.i.d.Z.21*-program and in FFF-protests. Based on the findings, results are discussed with the CCE literature prevailing, and, eventually, conclusions for future *k.i.d.Z.21* learning interventions and CCE, in general, are drawn.

## 2. Climate Change Awareness and the Knowledge-Action Gap in the Context of CCE

Ever since climate change has started to enter the global political agenda, the great potential of the education sector in dealing with the topic has been recognized and is still at the top of the agenda, referred to as CCE for sustainable development [10,19].

In fact, CCE can be one of the most effective means to shape and influence people's collective engagement in creating a climate-friendly society [1,20]. This requires people's awareness of climate change [20] of which educational attainment has been identified as the most important predictor. However, the term of CCA is used inconsistently in educational research due to its various definitions in literature [1,20]. It can comprise, for example, personal knowledge, interest, and attitudes [21–23].

In the early years of CCE, there used to be a tendency to distinguish between teaching knowledge and attitudes on the one hand and encouraging pro-environmental action on the other hand. In fact, there used to be disagreement about which of these disciplines should be focused on more [24]. Meanwhile, in the field of environmental education research, it is commonly accepted that cognitive and action-related factors interrelate with each other and that both have to be considered in CCE [25].

Yet, the gap between public knowledge and actual engagement remains troublesome and has also been a strong focus in the CCE research [26–29], often communicated as a knowledge–action gap [5]. It is caused by several affective and cognitive processes that, in turn, are influenced by multiple factors that together contribute to public (dis)engagement with climate change, including formal education [30–32].

As the determining factors and complex interactions of CCA and climate-friendly behavior still remain ambiguous [33], this study understands CCA in a broad sense. It relates to a hypothesized model of Teksoz et al. [34] that contains four factors of attitude, level of concern, and climate-friendly behavior, including multiplicative actions and knowledge, while all components are influencing each other. These are listed as follows:

*2.1. Attitude, Including a Sense of Responsibility, Self-Efficacy, and Locus of Control (LoC)*

Pro-environmental behavior can be influenced by the individual attitude towards taking action in the context of climate change. In turn, this comprises different factors; for example, a sense of responsibility towards climate-friendly behavior [35,36].

Additional components include, for example, self-efficacy [37]. This is also called a personal sense of competence [38], the estimation of successfully implementing someone's own competencies and actions when confronted with difficult situations [37]. It mutually enforces collective efficacy or a collective sense of competence, which is the belief in the ability to achieve goals while working together with a group [16].

Self-efficacy is also closely linked to LOC [39], which describes the degree of which an individual is convinced to be able to control phenomena, such as contribute to climate protection. In turn, external control is the extent to which the individual classifies the event as fate, circumstances, or controlled by "mighty others" outside their personal context of influence [40].

*2.2. Level of Concern*

Additionally, research shows, the more people feel themselves and others affected by climate change in their present or future lives, the more likely they will engage in climate-friendly actions [25,41]. Yet, psychological, geographical, and temporal distances of climate change prevent personal concern of people [42–44].

*2.3. Climate-Friendly Behavior*

Environmental education has the potential to initiate actions among a large proportion of the population, as well as reinforce existing pro-environmental behavior [24]. Although some environment-friendly actions are out of teenagers' scopes, for instance, choosing a green electricity provider or using electric car mobility, a reduction of CO2-emissions requires efforts from multiple societal sectors including households. Thus, it is important to encourage young adults to reduce their carbon footprint in their everyday life [9,41].

Moreover, multiplicative action is integrated here. Due to multiplier effects, families and communities benefit from individuals sharing what they have learned [10]. Here, teenagers can engage as change agents when influencing their family and friends by discussing climate-relevant topics or influencing them to act in a more climate-friendly manner [45].

*2.4. Knowledge*

Though not being the only influencing factor, knowledge about the effects and consequences of climate change is still a crucial precondition for acting in an environment-friendly manner [5,36,41].

## 3. Applied Democracy for Young Adults In-School and Out-of-School

*3.1. Democratic CCE Setting in Practice—The Case of k.i.d.Z.21*

Mochizuki and Bryan [10] understand CCE as "processes aimed at improving the degree to which an education system is prepared for, and is responsive to, the challenges of climate change".

Yet, current climate change education still tends to focus on scientific knowledge [8,46]. However, educators must rethink climate change mitigation and adaptation in ways that are not purely technical but also socially transformative [47]. Besides, preparing students for unknown or only partially understood futures with traditional learning approaches is seen as inadequate [48]. Yet, typical school environments today still reflect a dominance of traditional teaching styles with a technical understanding of learning while encompassing contents far off from students' real lives and personal interests [49,50].

Rather, teaching and learning approaches are required that empower students to be able to act in a rapidly changing, risky future [47]. Given the socio-scientific nature of climate change, CCE

demands learning approaches that are inquiry-based and participatory in supporting learners to develop competencies that are transferable to new, uncertain situations [51]. This is best addressed through approaches that fully allow students to explore the nature of the problem themselves while permitting them to discuss and debate collectively about appropriate pathways forward and to take positive actions [8]. Here, a self-guided, inquiry approach is considered as particularly appropriate [52] for preparing students for democratic participation in-school and out-of-school [53].

Addressing the gap of democratic learning CCE settings in practice [54], in 2012, the working group of Education and Communication for Sustainable Development of the University of Innsbruck initiated a climate change educational programme called *k.i.d.Z.21—competent into the future*. The project aims are to raise students' awareness about climate change and its consequences among young people and prepare them for the challenges of the 21st century by strengthening their awareness, including self-efficacy and their capacity for action [14,55,56]. Within *k.i.d.Z.21—competent into the future,* formal and informal learning settings are combined in five different modules (for the structure of the programme, see Figure 1). These modules are integrated into the curricula of high schools in Austria and Bavaria, addressing students at the age of 12–16 years on average. The transdisciplinary approach is first put into practice at the kick-off workshop. In plenary discussions with teachers and diverse stakeholders from science and the public, students are given a voice in topics about climate change. Moreover, students get motivated by switching from the classroom to an out-of-school setting and meeting participating peers of other institutions [57].

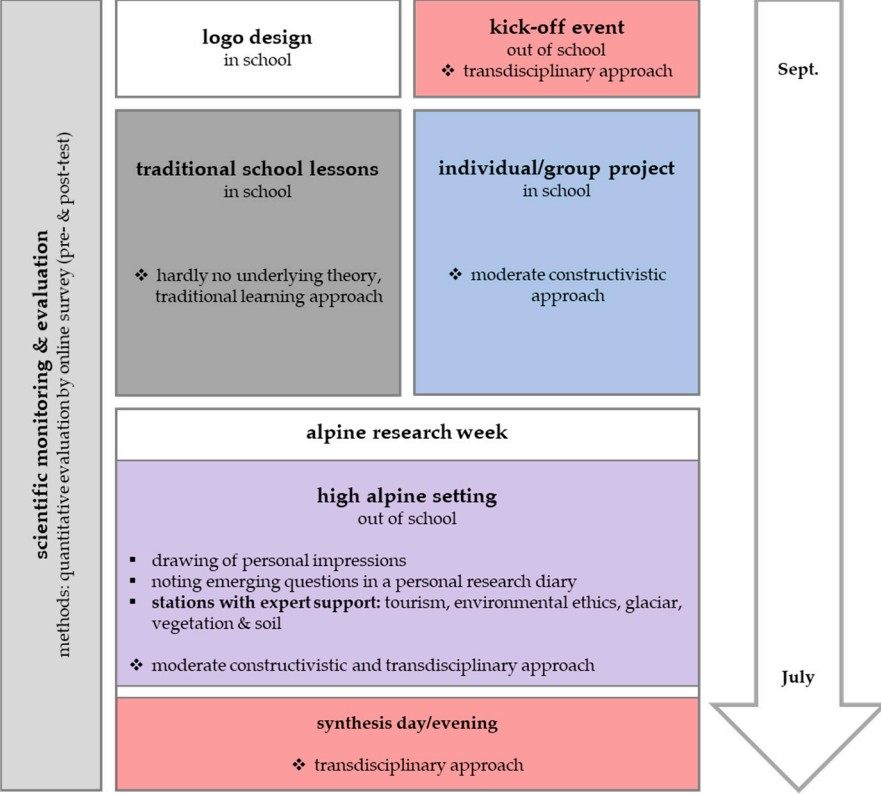

**Figure 1.** Overview of central modules of the *k.i.d.Z.21* project design per school year. Colors of modules reflect applied learning methods: grey (traditional teaching styles), blue (moderate constructivist approach), red (transdisciplinary approach), and purple (mix of moderate constructivist and transdisciplinary teaching styles in authentic learning setting). The figure is adapted following Lars Keller [14].

During an entire school year, the topic of climate change is dealt with in both teacher-centered classical school lessons and in individual research projects. The individual project modules grant a high degree of responsibility to the students, deciding about content and methods themselves.

They can elect among the discipline of natural sciences (e.g., mathematics, physics, chemistry and biology) or/and social sciences (e.g., history, arts, religious education). This highly autonomous way of learning is based on the principle of moderate constructivism, the idea of learning as an active process of knowledge construction within the individual mind with a minimum of instructions given [58]. Its goal is to enable learners to develop plausible interpretations of certain sectors of knowledge without or beyond the information given [59], respecting various perspectives, world views, and alternative interpretations [60]. This way, the students actively produce their own learning outcomes by using both their creative and analytical skills and make their own decisions in an interdisciplinary setting.

This is also in line with the requirements of climate change education, which demands a focus on the kind of learning, critical and creative thinking, as well as on capacity building that enables teenagers to engage with the information, in addition to inquiring, understanding and eventually taking what they determine as appropriate actions to respond to climate change [8].

Moderate constructivism can be applied particularly well in discovery- and inquiry-based learning settings [61–63]. Moreover, learning processes should be embedded in authentic learning contexts, where knowledge can be directly linked to old and new experiences by observation [64]. Thus, at the end of a school year, moderate constructivism is applied in the authentic surroundings of the high alpine mountains with visible indicators of climate change effects in the format of High Alpine Research week. There, accompanied by experts from social and natural sciences, students individually work on and respond to self-defined research questions. During the week, they pass through four different modules, such as tourism and environmental ethics from social sciences, as well as vegetation, soil, and glaciers from natural sciences. At the same time, research questions and results are continuously documented by students in their research diaries and are finally verified on the synthesis day. To sum up, *k.i.d.Z.21* attempts to link teenagers' real-life concerns and interests with scientific argumentation in both in-school and out-of-school learning settings while applying the learning principle of moderate constructivism.

In fulfilling the claims of ESD, scientific research has an important role in monitoring and evaluating learning settings [65]. Thus, each project year starts with a pre-test and ends with a respective post-test, both measuring students' climate change awareness, comprising the components defined in the previous chapter. Besides, students' perceived effectiveness of the learning intervention is evaluated. Here, by integrating the self-reflection of learning, the active participation of students is fostered [11]. These results, in turn, are incorporated in the modification of future *k.i.d.Z.21* learning settings following a transdisciplinary approach.

### 3.2. CEE in the Context of Fridays for Future

The current way of teaching about climate change has followed a largely apolitical cognitivist approach in the first place [54], as it is focused on an individual sphere of private mitigation action [66]. However, in the requirement of a societal transformation, climate-friendly actions are urgently required on a collective scale and should also involve active political citizenship. Thus, instead of terminating at the end of the lesson, education can provide a foundation for democratic engagement and action. This becomes especially relevant as political engagement is considered as being among the most effective actions for the environment, due to its outreach to mighty societal actors of business and government [16]. In order to enable collective action, Levinson 2012 [67] calls for explicitly questioning the sociopolitical relations between school and community [68]. Thus, classrooms should be world-oriented and encourage "out-of-the-box" thinking by opening for direct democratic participation of students outside schools [69,70].

As in most industrial countries, the legal right to vote is obtained at the age of 18, the options of young people for directly participating in democratic processes for shaping their sustainable future are minimized. Not willing to accept her apparent impotence, 16-year old student Greta Thunberg implemented her civil right to protest, which is now imitated by teenagers and other age groups in 148 countries worldwide and has become one of the major movements of civic climate activism as a response to missing political action towards climate change [71]. Focusing on Innsbruck, Austria,

weekly striking began on the first Friday of February 2019 and is still going on, while it culminated in its number of participants on the dates of global and national earth strikes (March, April, May, September, and November 2019) with the highest number of 20,000 participants (one-sixth of the city's overall inhabitants) [15]. Thus, after decades of political lethargy among young generations, the current activism in which adolescents take a stand for actively creating their future can be regarded as a positive development.

A potential influence of climate protests on young peoples' climate change awareness is assumed due to the group identity model of collective action [72]. According to this, participating in opinion-based groups, like FFF, contributes to form a collective identity. This, in turn, is reinforced by perceived collective efficacy, which interrelates with individual efficacy, a part of climate change awareness as defined above [17]. These two factors finally have the potential to initiate desired collective action towards climate protection [18].

Thus, the following chapter investigates if components of climate change awareness differ among the two groups of participants in FFF and non-participants after the learning intervention.

## 4. Research Design

### 4.1. Study Design

In order to analyze potential differentiating components of CCA among the two groups of participants in FFF and other students more in detail, in this study a mixed-methods approach is applied during two main research phases [73]. In the first phase, different components of climate change awareness as defined in previous chapters, are analyzed quantitatively by two standardized online questionnaires each before and after the learning intervention (pre- and post-test). They comprise mainly closed questions in the form of a six-level Likert scale, which is explained more in detail in the subsequent chapter and shown in the supplementary material.

Within the second phase, it is attempted to identify the potential sources of the differences in climate change awareness, whether in the CCE *k.i.d.Z.21* learning setting or/and in the FFF-climate protest in face-to-face interviews. Moreover, a more detailed understanding of the interrelations of both fields should be enabled.

The study period lasted one school year, starting with the pre-test in September 2018 before the climate change issue was really dealt with, and ending with the post-test in July 2019.

For conducting the studies with students, certificates of good standing are applied at the Board of Ethical Questions in Science of the University of Innsbruck and received on a regular base. The last approvals were addressed to the climate change education projects of CryoSoil_Transform (P7160020-020) and SEAS (P7160-026-018), which were received on 20 May 2019 and on 1 October 2019, confirming correspondence with all requirements of the ethical principles and guidelines of good scientific practice of the University of Innsbruck. All subjects gave their informed consent for inclusion before they participated in the study.

Some limitations of this study have to be noted. First, there was no control group queried to compare the results of the intervention with students who solely participated in traditional school lessons. Besides, by responding to the online questionnaires, students are only able to indicate their subjective perceptions. Moreover, personal interviews aimed at revealing additional information, which has been disclosed in quantitative analysis. However, these qualitative results are not representative and are presented in Table S2 in the supplementary material. Moreover, the possibility of several biases on the interviewer's and respondent's sides is given, such as the framing effect, the retrospection effect, or social desirability [74,75].

Additionally, this study does not aim to identify the direction of causality of participating in FFF and changing factors of climate change awareness, as in FFF, only certain types of characters participate. Rather, it conducts a comparison between the two groups of participants in FFF and other students who solely participated in *k.i.d.Z.21* in order to reveal components of climate change awareness that are possibly interesting for further research.

Besides, before the start of the program, much time was invested in training teachers about the concept of *k.i.d.Z.21*. However, the design of respective traditional school lessons was within their own responsibility with the condition of integrating the topic of climate change in their lessons. Classical school modules themselves were not monitored and evaluated so they might differ in their content and teaching approach. Further, the time students can spend on their projects during school lessons depends on each teacher/school.

*4.2. Methods*

Both questionnaires, pre- and post-test, were scientifically developed and validated during a continuous period, as they have been part of the *k.i.d.Z.21* project since 2012/2013 and have been applied every year since then. During the pilot phase, online questionnaires were sent out to the secondary school of Tyrol, addressing the respective age group that was not part of *the k.i.d.Z.21* school network. During the project period, the questionnaire was continuously modified while the last modification was in 2016.

They comprise introductory questions on individuals' life satisfaction and personal views of the future, which are not part of the current study, in addition to selected components of climate change awareness including knowledge, attitude, personal concern, climate-friendly behavior, and multiplicative actions.

The post-test of the school year 2018/2019 was additionally complemented by a few quantitative questions relating to FFF, such as asking for participation ("yes" or "no"). The reason for this was the arising climate protests in Austria beginning in February 2019 and the assumed influence of the movement on students' climate change awareness.

The post-test is further complemented by questions evaluating the learning intervention, referring to single project modules. Both questionnaires are developed and implemented in a questionnaire software and evaluated quantitatively by IBM SPSS Statistics 25.

For the quantitative analysis of the sample, the *t*-test for dependent samples is deployed in order to compare the results of the pre- and post-test, while significant changes are indicated by the symbol "*". All results are presented in Table S1 in the supplementary material. The *t*-test was chosen as the sample size is large enough to assume normal distribution of the data. The evaluation is based on a level of confidence of 95% ($z = 1.96$). A significance level of $p < 0.5$ is considered as significant in the analysis of the data. In order to measure the practical relevance of the results, the correlation coefficient Pearson's *r* according to Bravais–Person is calculated. The interpretation of the effect size (r = 0.10 weak effect, r = 0.30 medium effect, r = 0.50 strong effect) is used according to Cohen (1992) [76]. Measuring the scale reliability of the constructs composed of different items of the questionnaire, Cronbach's alpha was calculated. The interpretation of Cronbach's alpha is used according to Mohammed and Pandhiani [77] in a study about students' evaluation of teaching effectiveness (a value of Cronbach's alpha >0,7 is accepted to be sufficient).

Secondly, personal interviews with selected students from phase 1 who also took part in FFF enabled a better understanding of the role of FFF for the learning intervention. These 20-minutes semi-structured interviews were conducted individually just before the post-test with ten voluntary students (five female, five male) of Karl-von-Closen High-School of Eggenfelden (KvC) during the High Alpine Research Week. These teenagers were selected on a voluntary basis due to the criteria of having participated in FFF. The interview protocol examined the effects of participating in FFF on selected components of teenagers' climate change awareness (e.g., self-efficacy and climate-friendly behavior). Besides, the additional value of participating in FFF compared to *k.i.d.Z.21* was asked. The interviews were recorded by digital recording and were then fully transcribed [78]. Consequently, each transcript was analyzed by structured content analysis [79,80], for the attitude and patterns of argumentation, using the software MAXQDA 2018. The interviews were conducted in German and selected answers were translated into English by the authors of the study.

The participants of the study were not chosen randomly. They had already been part of the *k.i.d.Z.21* network in the school year 2018/2019. During the project period, different school types from various regions participated: Four secondary schools from rural and urban Tyrol with students between 13 and 16 years old, one middle school from rural lower Austria with students between 11 and 13 years, and two secondary schools from rural southern Germany with students between 13 and 15 years. In addition to secondary and middle schools, one higher federal research institute for nutrition, food- and biotechnology was involved. In total, N = 187 students took part in the pre-test, while N = 180 students, at the age of 11–16 years, completed the post-test. Due to missing or wrong entered personal codes, which the students had to create the first time during the pre-test, the final sample size was reduced to N = 169. The codes permit both to guarantee anonymity for the participants, and at the same time, to evaluate the intervention, comparing students' answers between the pre-and the post-test. Of the participating students, 60.9% were female, while only 39.1% of the participants were male. In total, 53 of the students took part in the FFF movement, 56.6% of those were female, while 43.4% were male students.

## 5. Results

*5.1. Quantitative and Qualitative Results of Changed Climate Change Awareness*

In the results section, quantitative and qualitative findings together are demonstrated per each construct of CCA. The overall sample size is N = 169. A group comparison is demonstrated, for which the overall sample is divided into students who took part in the FFF movement (N = 53) and other students (OS) who did not participate in FFF (N = 116). The quantitative results are shown in Table S1. Subsequently, the qualitative answers of the interviews with N = 10 students are presented in more detail.

### 5.1.1. Attitude Including a Sense of Responsibility, Self-Efficacy, and LoC

The construct Responsibility was divided into responsibility for climate-friendly behavior of oneself and of others, while the latter one is composed of five different items (see Table S1). Although the means of both types were raised after the project period in both groups, the changes were not significant for the whole population. Yet, the responsibility of FFF participants was higher than that of non-participants both before and after the project. These findings again appear in the interviews when asking about the perceived effects of participating in FFF in general, as it is also shown in Figure 2.

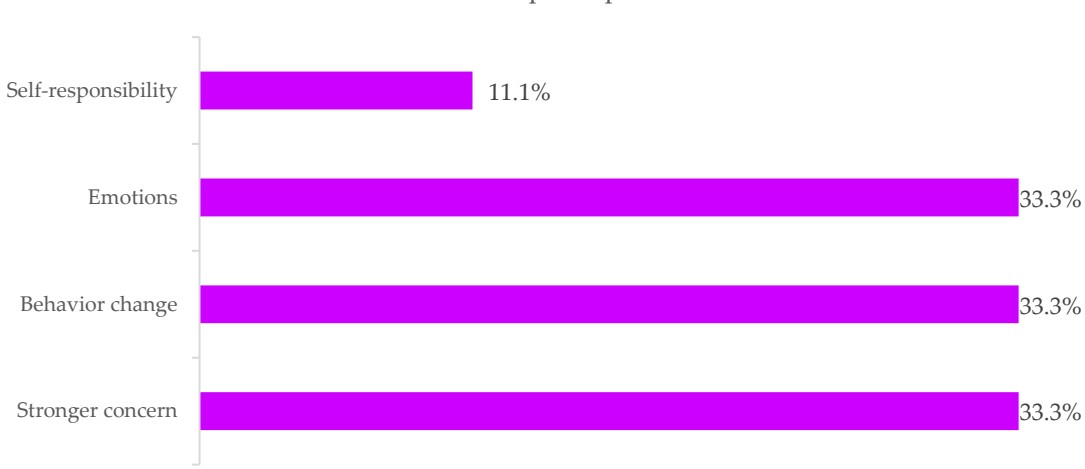

**Figure 2.** Quantity of codes categorized when asking "What effects did participation in FFF have on you?"; results of personal interviews, N = 10. *Self-Efficacy.*

Besides, after the learning intervention, both groups showed a significant increase in self-efficacy. Although the mean change was higher among the FFF participants, the mean values were higher among the non-participants after the learning intervention. When asking about the general perceived effects of participating in FFF in the interviews, as shown in Figure 2, effects on self-efficacy were not explicitly mentioned. However, when questioning in a more specific way about perceived effects in their own ability to act climate-friendly, 90% of students confirmed a respective increase triggered by FFF. Particularly, this was effected by a perceived collective self-efficacy that was transmitted by the crowd of people, which together with other results, is illustrated by selected students' statements in Figure 3.

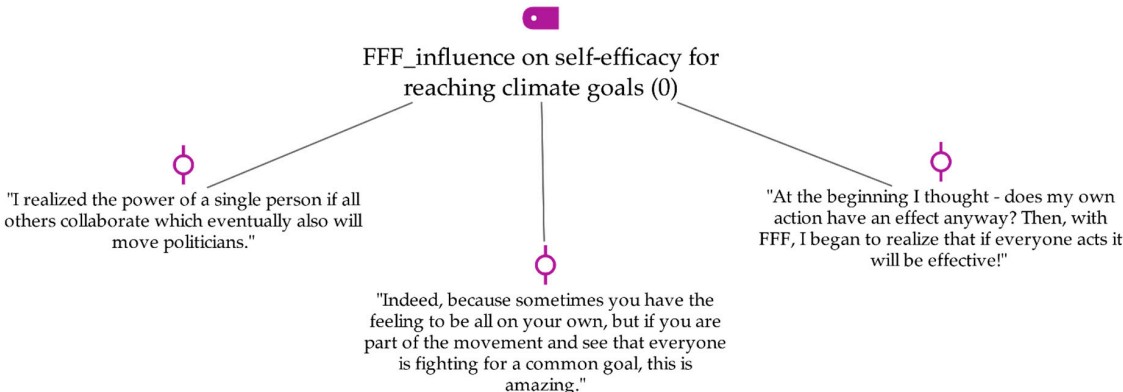

**Figure 3.** Selected quotations from interviewed students asking for self-efficacy "How did participation in FFF affect your feeling of being able to act climate-friendly?". N = 10.

### 5.1.2. Locus of Control (LoC)

LoC was measured by three items (see Table S1) and increased significantly among both groups. Before and after the project, the mean value was higher among FFF participants. As self-efficacy is closely linked to LoC, the qualitative findings of the paragraph above can also be applied in this section.

### 5.1.3. Level of Concern

Level of Concern was measured by eight different items categorized in concern for one's own life, concern for the life of one's family, and concern for others. These subcategories were further separated into two time-scales of present and future (today and in 20 years). While in both groups, concern is raised after the project period, generally, values are higher for other people than for oneself and increase along with greater time distance (e.g., Table S1). Comparing the FFF group with non-participants, the overall construct of concern was higher among FFF-participants than among non-participants. However, the values were only significantly higher for the categories of "my life today", "the life of my family today", and for "others today", as well as for the subcategory "other people worldwide in 20 years". Considering all categories, there is an increase in concern from the beginning to the end of the learning intervention. However, interestingly, only the results for the future time scale (in 20 years) are significant. Generally, it shows that ratings of the students for today are smaller than the ratings for 20 years in the future and that of the others in 20 years. This also becomes evident when asking about the perceived effects of participating in the climate protests in the interviews, again shown in Figure 2. One-third of FFF participants claimed that they became more aware about the dimensions of climate change and that they felt themselves more heavily concerned about this topic, which is also highlighted by the following citation:

> "There, I finally began to think about what climate change really means, it's getting warmer and soon, there will be no snow anymore. That shifted my awareness completely".
>
> student FF of KvC Eggenfelden [81]

Additionally, three students mentioned incentives that were transmitted by attending FFF, e.g., "it's the best incentive for personal behavior change and I will keep on acting climate-friendly at home" [82]. Other results of behavioral change are presented in the respective paragraph.

### 5.1.4. Behavior

Climate-friendly behavior was queried by various items comprising specific activities in a person's everyday life. These were divided into nine categories of different action areas: active information seeking, multiplicative behavior, engagement in climate change issues, energy and water consumption, waste separation, alimentation, and consumption behavior in general (see Table S1). While values raised significantly for the whole sample, when separating between groups, the sub-item of alimentation only changed significantly among FFF participants. Yet, in both groups, mean behavioral change among FFF-participants rose significantly, and was higher in engagement and multiplicative actions. Additionally, in every category, values were higher among the FFF-group after the learning intervention, while the highest values among FFF-participants were detected in the categories energy consumption and consumption behavior. The lowest change was in the categories of energy and water consumption, as well as in waste separation. In both groups, the lowest values belong to the sub-item "I also join information events about climate change".

In the interviews, as illustrated in Figure 2, one-third of students already mention effects on their behavior when asked for them openly. When questioning them directly about the types of behavioral change in detail, mainly consumption behavior is mentioned by 88.9% of stuents.

Specifying types of behavioral changes, Figure 4 illustrates that the majority think twice before buying a product (62,5%), whereas more than a third have begun to reduce plastic since participating in FFF. Additionally, one quarter is more conscious of conducting their everyday actions in a climate-friendly manner. Besides, 78% confirmed enhanced multiplicative actions, and more than half of the students claimed to have changed their mobility behavior by refusing car rides offered by their parents and instead switching to public transport or bikes. Two students also claimed active information seeking since participating in FFF and the refusal of meat.

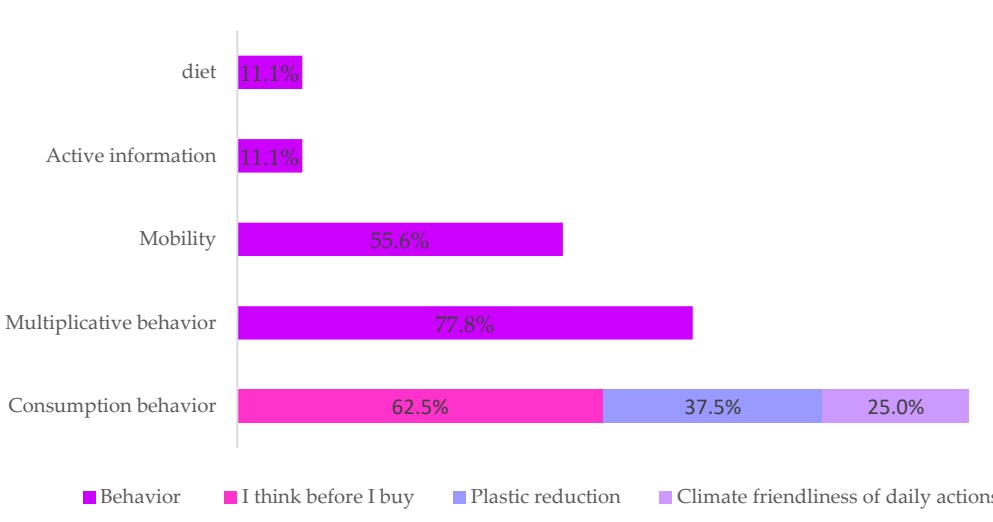

**Figure 4.** Quantity of main codes in different behavioral categories and sub-categories. Question: "How did participation in FFF effect your behavior concerning climate change?"; N = 10.

### 5.1.5. Knowledge

Students had to rate on a 6-level Likert scale if statements were true or false. Here, questions were selected that relate to the modules (tourism, environmental ethics, vegetation/soil, glaciers) the students completed during Alpine Research Week. While in both categories (right and false answers)

knowledge had significantly risen in both groups, among FFF participants, knowledge values were higher than among those who did not participate. In the interviews, no increase of knowledge was mentioned by the students since starting to participate in FFF, as this, in general, is a difficult aspect to assess by themselves.

### 5.2. Qualitative Results of Added Value of FFF Compared to School/k.i.d.Z.21

In order to enable a better understanding of the effects received by participating in FFF compared to *k.i.d.Z.21*, students are additionally asked in personal interviews if they had noticed an added value of FFF compared to the CCE learning intervention.

As illustrated by students' statements in Figure 5 above relating to FFF, the majority of codes claimed an increased self-efficacy. Here, in general, FFF was considered to enhance the feeling of getting into action and taking concrete actions rather than the *k.i.d.Z.21* project, which was considered more adequate to provide an informational background or to raise motivation. Besides, the educational intervention is regarded as valuable complementation to FFF, "which should not be foregone" [83]. Moreover, other components of climate change awareness were enhanced, such as self-responsibility and the level of concern. Effecting also a stronger commitment, FFF even raised an additional component of awareness. Two of the interviewed students, who were also part of the local FFF-organizing team, additionally initiated project activities beyond the learning intervention, such as zero-waste workshops and do-it-yourself plastic-free cosmetics [84]. A student project that was created within the scope of FFF and within *k.i.d.Z.21 individual project module* is "*bike for future*". In the form of a biking competition with a sustainable price-winning option, a group of students incentivized their classmates to cycle to school for a defined period of 21 days in total. Related project activities were sponsoring acquisition and creating communication material, including a project website and printed flyers. Besides, students had to get into contact with public stakeholders, like the city council or the police office, when asking for permission. Both initiators and participants of this project then took part in FFF with "ringing our biking bells instead of clapping our hands [84]". Moreover, a student commented "without FFF, the individual *k.i.d.Z.21* project had only turned out to be an ordinary presentation" [84].

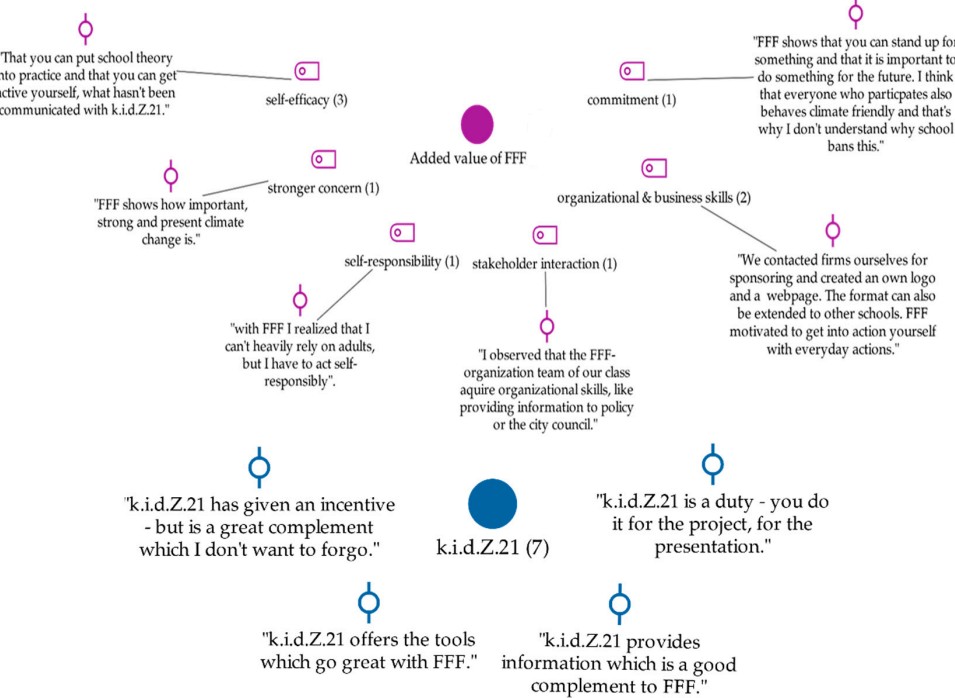

**Figure 5.** Selected quotations from interviewed students and derived categories from the question: "What do you learn by participating in FFF compared to *k.i.d.Z.21*?" Different colors indicate source of learning; purple = FFF, blue = *k.i.d.Z.21*; N = 10.

## 6. Discussion

The findings of this study should be considered in light of some limitations arising from a lack of several potential explanatory variables that influence an individual's environmental attitudes. As confirmed by current research, apart from the school context, environmental attitudes of adolescents result from a complex interplay of additional social factors, such as family background and interactions with peers [57]. As the applied online questionnaire in this study does not control for these additional influencing factors, the significant mean changes of components of climate change awareness cannot be unambiguously attributed to the *k.i.d.Z.21* learning intervention. However, this study endeavors to compensate for this with a highly diversified sample comprising of different school types, school regions, and age groups of students.

Besides, the origin of higher means of some constructs among FFF participants cannot be clearly attributed to the movement itself, as they might be an inherent part of individual character and thus, might have already existed.

Considering these limitations, with the exception of responsibility, all residual items of climate change awareness could be enhanced by the learning intervention of *k.i.d.Z.21* in both groups. With regard to the construct of attitude, previous studies have already shown that it is quite a difficult aspect to change [57].

The higher level of individual concern among FFF participants is especially relevant since this is a critical factor for environmental-friendly action [44]. This is also evidenced in van Zomeren's theory of collective identity, where perceived injustice, a very similar factor to individual concern, influences group identification and collective action in the last instance [17,72].

Although FFF appears to partly overcome forms of psychological distance, spatial and temporal distances still persist in single items of the construct. Climate change is perceived to have less impact in Europe today and in the future compared with its impact on personal and global levels. This has already been proven several times in literature, as individuals in Western countries imagine that climate change will mostly affect faraway nations [44,85]. Sadly, even after completing the learning intervention, where students were even surrounded by visible climate change indicators in the figure of reduced glaciated area during High Alpine Research week, their responses still suggested that they felt that climate change was not a domestic issue, but something affecting people on the other side of their world. Possibly, students were not aware that they actually live on the European continent. Our future programmes should address this gap by focusing on local climate change impacts in additional project modules, perhaps in individual student projects. The level of concern has possibly risen due to participating in FFF and is a factor that ought to be a major focus in future research. This becomes even more evident when asking students openly about the perceived effects of participating in FFF, as a higher level of concern is also mentioned here.

Another significant difference among FFF-participants is revealed in alimentation, represented by the item of refusing meat. This is interesting as diet is a difficult field to change. It is recognized as being a classic case of a knowledge–action gap in the context of environmental-friendly behavior [86–88]. As previous studies show, willingness to refuse meat in favor of the environment is usually also low among young people [25,86,89]. However, as meat production yields the highest carbon emissions [90], collective action from consumers is especially important in this field.

Differences in residual sub-items of climate-friendly behavior were noted where students obtained values that significantly changed in buying energy-efficient products. Here, possible reasons are that the FFF group focused more on product sufficiency than buying new material. Interestingly, other students also had significant values with climate-friendly transport compared with the FFF participant results showing no significant changes. Yet, values of FFF participants were already were quite high even before the learning intervention. This, however, contradicts the findings of qualitative interviews, where half of the codes of behavior changes stated that, following participating in FFF, they had begun to cycle more often. However, the interviews only refer to a small sample from a school in the countryside of lower Bavaria where quite long bicycle distances had to be travelled (approximately eight miles).

Regarding the whole sample, the result is very positive, as young peoples' willingness to change their choices of modes of transport has also been quite low in the past [25].

While not being a significant change in quantitative research, but appearing in qualitative results, multiplicative behavior can possibly be enhanced by FFF participants as they might also influence their peers to join them. This is especially relevant regarding peer effects as additional influencing factors of climate change awareness [57]. Hence, a democratic education context allows students to act as change agents and thus, can multiply collective action for a climate- friendly society [45].

The factor of climate friendly engagement, which was significant in both groups but was double in size among FFF participants, is very likely to have risen as a result of FFF as one of the most easily accessible political platforms with which young people can currently engage. This is especially relevant in the face of high impact of collective political actions for climate protection. With their vast outreach, they obtain the potential to move major societal actors such as business and government on local, national, and international levels [16].

With no value changes of energy, water consumption, and waste separation between both groups, it can be assumed that these actions were already commonly applied within the students' family households in Austria and southern Germany before the learning intervention. These results are also in line with those obtained by prior studies in the UK that measured young people's willingness to undertake climate-friendly actions, with major findings in energy consumption and recycling [25]. These changes in behavior can be attributed not only due to climate protection reasons, but also to cost-saving efforts.

Continuing with the theme of constructs that showed significant changes in both groups in the quantitative interviews, but were higher within FFF participants, an increased sense of individual self-efficacy can be clearly identified among the interviewed FFF participants. This could be also traced back to the different forms of questioning self-efficacy in the personal interviews compared with the online questionnaires. Firstly, in the face-to-face interviews, the questioning was more detailed and specific. Secondly, the concept of self-efficacy might have been addressed via a leading question, for example ("How did participation in FFF affect your feeling of being able to act in a climate-friendly way") which was further addressed to concrete climate-friendly actions. The online questionnaire inquiry was more subjective, with regards to more holistic climate change, asking ("How well do you feel prepared towards the changes induced by climate change?"). Here, the limits of ambiguous causality must also be considered since people are more likely to get engaged politically if they already obtain a personal sense of competence, as well as a belief in their collective competence [16]. Nevertheless, FFF-participants' perceived increased sense of self-efficacy is in line with the hypothesis of enhanced collective efficacy resulting from the theory of group identity [17,72,91].

Knowledge is a further construct with significant changes in both groups, but higher ones among FFF participants. In the context of climate change, knowledge is highly uncertain, shortly-lived, and context-specific [27,92]. On the other hand, in past research, extensive knowledge gaps have been identified on both student and teacher sides due to misconceptions of climate change [93–95]. In order to bridge these gaps of both parties, it is recommended that a high degree of learning responsibility is allocated to students, which again can be facilitated by the approaches of inquiry-based learning and moderate constructivism [91].

As was already stated in the first section of the study, the importance of knowledge in the face of knowledge–action gaps has to be considered critically, as individual behavior is influenced by a plethora of additional influencing factors [96]. Nevertheless, knowledge is still regarded as a necessary content of CCE [25].

Beyond examined components of CCA, interviews also revealed enhanced motivation that was brought about by attending FFF. Apparently, climate change engagement helps to overcome feelings of distress, hopelessness, and fear [23]. In turn, motivation is an important factor for climate-friendly action [33], and thus, attending FFF can be attributed to being beneficial.

Summing up, *k.i.d.Z.21* participants who also took part in FFF show higher means in action-related components of CCA. For the CCE *k.i.d.Z.21* setting this means the format of the individual project module that is based on moderate constructivism appears to have a high potential of triggering action, particularly when linked to FFF. As students' answers in connection with the "*bike for future*" initiative show, this combination leads to positive effects beyond raising CCA-components, such as additional practical skills that are useful for collective action towards climate protection.

Thus, the combination of learning modules based on democratic learning principles with applying direct democracy out-of-school seems especially fruitful for enhancing climate-friendly action.

## 7. Conclusions

The central aim of this study is to investigate if single components of CCA differentiate among *k.i.d.Z.21*-participants who have and who have not been involved in FFF-protests.

Indeed, higher mean changes and higher mean values of single components of CCA and related climate-friendly action were mainly found among FFF-participants. Thus, the findings are in line with the previous theory of enhanced self-efficacy and subsequent collective action when participating in protests [17] and also apply within the context of moderate constructivist CCE interventions.

To confirm these findings, the differentiation between protest and non-protest *k.i.d.Z.21*-particpants will continue in future scientific monitoring of *k.i.d.Z.21* learning interventions.

Implications from these findings for designing future CCE learning settings aiming at bridging the action gap are that linking protest participation with individual project modules based on moderate-constructivist democratic learning approaches appears to be rewarding.

Beyond raising CCA, this combination also exploits the potential to create additional action-related skills that appear to be useful for climate protection. To confirm this hypothesis, further individual projects will have to be examined in connection with the FFF-protest action in more detail. For future research, it can be interesting to evaluate whether other *k.i.d.Z.21* modules also fulfill the potential for similar outcomes. Scientific monitoring focusing on students' evaluations of CCE learning modules helps to integrate and/ or further develop democratic elements within different learning modules.

In conclusion: Combining both FFF-protest participation and CCE can be mutually beneficial. Action-related components of students' CCA are strengthened, student motivation increases, and practical skills are developed. All in all, the recommendation for education policymakers and practitioners can only be to support teenagers to exercise their democratic rights within in-school and out-of-school settings. This can be a strong contribution to climate change learning and action.

**Supplementary Materials:** The following are available online at http://www.mdpi.com/2071-1050/12/5/1748/s1. Quantitative and qualitative questionnaires hat are not shown in the article are provided here as tables: Table S1: Quantitative questionnaire and results and Table S2: Qualitative Interviews_Eggenfelden. Also Figures will be provided which have been already presented in the article: Figures 1–5.

**Author Contributions:** The four authors are all part of the working group of Education and Communication for Sustainable Development of the University of Innsbruck which is lead by J.S. The data of this article were gained within the scope of the climate change education initiative of *k.i.d.Z.21- competent into the future*, which is run since 2012 and supervised by L.K., while currently executed and administered by S.K. and V.D. which also contains management activities to maintain research data. Through the years, J.S. and L.K. have coordinated and maintained the funding sources of this initiative containing several projects and specific funding bodies, while it is currently co-funded by two projects comprising cryo-soil-Transform, funded by the Austrian Academy of Sciences (ÖAW) and by the project SEAS which is funded by the European Union. The design of the study concept and the overarching research goals was developed by S.K. and V.D. It is based on and complements the online questionnaire that has been developed by previous employees of the working group. After a short period, the concept was refined in cooperation with L.K. S.K. conducted writing of the *k.i.d.Z.21* project presentation including adaptation of Figure 1, as well as description of methodology and presentation of quantitative results including respective Table S1. L.K. also contributed to 3.1, to 4.1 and to parts of introduction and the conclusion, whereas V.D. was responsible for the residual written and graphical parts and prepared the paper draft. S.K. and L.K. also reviewed the article, as well as a third person outside the working group, while V.D. edited all their comments. The methodology of quantitative already prevailed, since the online questionnaire was already developed in 2016 by the previous team of climate education containing general constructs of life satisfaction and climate change related attitudes, knowledge, personal concern and behavior. This questionnaire was also disseminated to the prevailing

school network in October 2018 and July 2019, while only single constructs and items were selected for this article. S.K. analyzed online questionnaires with SPSS Statistics. The evaluation is based on a level of confidence of 95% (z = 1.96). A significance level of $p < 0.5$ is considered as significant in the analysis of the data. In order to measure the practical relevance of the results, the correlation coefficient Pearson's r according to Bravais-Person is calculated. The interpretation of the effect size (r = 0.10 weak effect, r = 0.30 medium effect, r = 0.50 strong effect) is used according to Cohen (1992) [76]. Measuring the scale reliability of the constructs composed of different items of the questionnaire, Cronbachs Alpha was calculated. The interpretation of Cronbachs Alpha is used according to Mohammed & Pandhiani (2017) [77] in a study about students' evaluation of teaching effectiveness (a value of Cronbachs Alpha > 0.7 is accepted to be sufficient). The qualitative approach was developed by V.D., who conceptualized the interview questions which were conducted in July 2019 in Obergurgl during the High Alpine Research Week 2019. Subsequently, they were transcribed and analyzed by structured content analysis using MaxQData, whereof the intercoder reliability was tested with a correlation coefficient of 80%. All authors have read and agreed to the published version of the manuscript.

**Acknowledgments:** Our great thank goes to diverse funding bodies that made the *k.i.d.Z.21* project possible, in particular during school year 2018/2019, the base of our study. These include the Land of Tyrol as main funding body of the project, as well as the Austrian Academy of Sciences (ÖAW) and the European Union (EU) as funding authorities of the PhD positions of first (V.D.) and second author (S.K.). Besides, we'd like to thank the University of Innsbruck and its Institute of Geography for funding publish processing costs. Many thanks also to our dear former and current colleagues of our working group who have done great work in advance and support us with their experience. Not to forget and especially, we would like to thank all the students from Austria and Bavaria who answered our questions in personal and digital form with their utmost patience and diligence. Eventually, thanks to all the teachers, who are motivated to participate in the *k.i.d.Z.21* project.

**Conflicts of Interest:** The authors declare no conflict of interest. The funders had no role in the design of the study, whether in the collection, analyses, or interpretation of data, nor in the writing of the manuscript, or in the decision to publish the results.

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
