# Peer review of "Bridging the Action Gap by Democratizing Climate Change Education—The Case of k.i.d.Z.21 in the Context of Fridays for Future"

_sustainability, doi:10.3390/su12051748_

Round 1

Reviewer 1 Report

The authors presented the supplementary file but not properly the type of questionnaires they have presented to the students.

I do think the type of questions are important to judge the different responses and final conclusions they took out of the study. Showing all this information could give a much cleaner presentation.

Reviewer 2 Report

The theme addressed in this manuscript constitutes a topic of global interest at this time. Awareness of climate change is an educational issue, although, in order to have a deep understanding of it, one would have to move away from certain ideological and political connotations, which unfortunately seems to determine thinking.

The initiative presented in this manuscript is brilliant, however, there are aspects of the writing that are not clear, making understanding difficult.

-The summary should include concrete results of the research carried out, as well as essential aspects of the methodology, about what has been done and who were the participants (youth, students, how old they were etc ...)

- The different sections of the Introduction focus on the problem of climate change, but it does not focus on how it can be addressed in the educational stage (from young children to university students).

-The age of the participants and the ideology of the family are aspects of great importance in the results obtained. The questionnaires address issues that not only involve education through this initiative, but would be influenced by the family and social context of the students. It is important that the authors comment on this, and may be an important aspect in the study. Did you follow up? Did the academic results correlate with the answers to the questionnaires provided in this research?

-In statistical analysis should appear in an independent section to 4.1 Study design and methods. In addition, it must appear after 4.2 Sample.

-It would be convenient to include the questionnaires used as Annexes. Similarly, know if they were previously validated, and in this case, the results obtained.

-Conclusion: It must be less extensive and more concrete. The conclusions must specify the results obtained from the objectives that were raised at the beginning of the investigation.

Round 2

Reviewer 2 Report

The article has improved in its writing, although there are still grammatical errors that should be reviewed.

Figure 5 does not present a suitable form.

Comment: The conclusion has improved, however, opinion phrases continue to appear, which are not obtained directly from the results achieved.

Example: “Moreover, combining democratic protest participation with autonomous CCE project modules seems particularly fruitful and beneficial in multiple ways”.

This idea should be discussed in the discussion. It is an idea powerful enough to put it as the conclusion of the study.

Comment: The study cannot end with such a phrase. "Last but not least, it should be stated that participating in FFF protests can also present an important stage for democratizing CCE."

Author Response

Response to Reviewer 2 Comments

Thank you again for your constructive criticism that helped me a lot to clarify the structure and results of my study.

Comment 1: The article has improved in its writing, although there are still grammatical errors that should be reviewed

Response 1: English grammatical style has been corrected and brushed up by intensified English prove reading of third and fourth author which is indicated by the change tracker in the word document.

Comment 2: Figure 5 does not present a suitable form.

Response 2: Figure 5 was corrected and integrated into text format.

Comment 3: The conclusion has improved, however, opinion phrases continue to appear, which are not obtained directly from the results achieved. Example: “Moreover, combining democratic protest participation with autonomous CCE project modules seems particularly fruitful and beneficial in multiple ways”. This idea should be discussed in the discussion. It is an idea powerful enough to put it as the conclusion of the study.

Response 3: The criticized aspect of Comment 3 was considered more extensively in several sections, like in results, discussion and conclusion that are all indicated in the tex in red:

Results

A students’ project that was both created within the scope of FFF and within k.i.d.Z.21 individual project module is called bike for future. In the form of a biking competition with a sustainable price-winning option, a group of students incentivized its classmates to bicycle to school for a defined period of 21 days in total.

Related project activities were sponsoring acquisition and creating communication material, including an own project website and printed flyers. Besides, students had to get into contact with public stakeholders, like the city council or the police office, when asking for permission. Both initiators and participants of this project then took part in FFF with “ringing our biking bells instead of clapping our hands [84]”. Moreover, a student commented “without FFF the individual k.i.d.Z.21 project had only turned out into an ordinary presentation” [84].

Discussion

Summing up, k.i.d.Z.21 participants who also took part in FFF show higher means in action-related components of CCA. For the CCE k.i.d.Z.21 setting this means: the format of the individual project module that is based on moderate constructivism appears to have a high potential of triggering action, particularly when linked to FFF. As students’ answers in connection with the bike for future initiative show, this combination leads to positive effects beyond raising CCA-components, like additional practical skills that are useful for  collective action towards climate protection. 

Thus, the combination of learning modules based on democratic learning principles with applying direct democracy out-of-school seems especially fruitful for enhancing climate-friendly action.

Conclusion

Implications from these findings for designing future CCE learning settings aiming at bridging the action gap are that linking protest participation with individual project modules based on moderate-constructivist democratic learning approaches appears to be rewarding.

Comment 4: The study cannot end with such a phrase. "Last but not least, it should be stated that participating in FFF protests can also present an important stage for democratizing CCE."

Response 4: The conclusion is now more focused on the central aims that are also reflected by in the last sentence which has been modified.
